# Optical Bubble Microflow Meter for Continuous Measurements in a Closed System

Michał Rosiak [ID], Bartłomiej Stanisławski and Mariusz Kaczmarek *[ID]

Faculty of Mechatronics, Kazimierz Wielki University, Kopernika 1, 85-074 Bydgoszcz, Poland;
mros@ukw.edu.pl (M.R.); bartlomiej.stanislawski@piekus.com.pl (B.S.)
* Correspondence: mkk@ukw.edu.pl

**Abstract:** This paper describes the design, operation and test results of a simple microprocessor-based device for measuring slow liquid flows. The device uses a module of 30 digital optical sensors to track the movement of a single air bubble inserted into a tube of flowing liquid. During a measurement session, the air bubble remains within the sensor module at all times, allowing the instrument to take measurements for any length of time. The liquid whose flow rate is being measured moves only in the closed tube system, without contact with other components of the device. The test of the device itself was carried out using a tube with an inner diameter of less than 1 mm, where the device is capable of measuring flow rates on the order of microliters per minute. Tests of the device showed good agreement between the measured volumetric flow rate and the reference flow rates of the infusion pump over the entire measurement range. The advantages and limitations of the device are discussed, as well as the prospects for developing the method.

**Keywords:** optical bubble tracking; flow rate detection; microflow; continuous measurement; closed system

## 1. Introduction

The need for precise measurement of slow liquid flows exists, for example, in cases such as measurement of the absolute blood flow in veins or flow through a peripheral organs [1], measurement of flow of lymph in lymph vessels [2], quantitative evaluation of the equipment for the target controlled infusion or drug infusion systems [3–5], flow rate measurement for calibration of microchannels used in liquid dosing applications [6,7], and testing of high-precision microfluidic flow controllers [8]. In addition to the appropriate sensitivity and accuracy of microflow meters, an important requirement that may be imposed on such devices, for example, in the context of clinical measurements, is the need for continuous or long-term monitoring of the flow process depending on other factors affecting the flow rate and measurement in a closed system, i.e., without contact of the liquid with the elements of the measuring apparatus. Literature studies show that the term slow flow, as well as the name microflow meter, is not unambiguous and cover the range of volumetric flow rates from milliliters, through microliters to nanoliters per minute. In this work, we will focus on a solution that measures flows with flow rates of microliters per minute, which are of interest, for example, in medical diagnostics and therapy.

Despite the needs noted above for measuring slow liquid velocities, the availability of appropriate measuring devices, usually called microflow meters, is limited, and the commercial devices offered usually measure flow rates of the order of milliliters per minute and often have properties that eliminate them from the above-mentioned applications (e.g., contact of sensors with liquids, heating of liquids). At the same time, it is worth noting that the enormous progress in the development of sensor and microprocessor technology significantly facilitates the construction of simple and inexpensive equipment whose capabilities can be comparable to the parameters of specialized and expensive research equipment.

Among the methods that are used for the measurement range of microliters per minute in the literature can be found gravimetric methods (measuring the mass of liquid on a balance) [9], mechanical methods (e.g., pressure gradient or cantilever methods) [10], thermal methods (e.g., calorimetric or time-of-flight methods) [5], optical methods (e.g., optical Doppler, particle image velocimetry) [11], and electromagnetic methods [8]. There are also methods in which two phenomena, e.g., mechanical and optical, are used simultaneously to measure the volumetric flow rate, as in the bubble method [2,3], in which the motion of a convectively entrained liquid bubble is tracked using optical sensors, or the cantilever method [12], in which the deflection of a beam under the influence of flowing liquid is recorded using, for example, a piezoelectric sensor. These measurement methods may differ in measurement range, limitations, complexity, or accuracy, and the choice of method should be related to the specific application.

This paper describes a simple device enabling precise measurements of the rate of slow liquid flow, which can be carried out in a closed system and continuously, i.e., from the moment of starting until interrupted by the operator. The method of tracking by optical sensors the movement of a bubble moving in a tube with liquid was used. The microcontroller receives signals from optical sensors detecting the presence of a bubble and controls the pinch valves that change the direction of bubble movement on the section of the tube located in the sensors module. The solution uses digital optical sensors that are not in motion, unlike the analog solution presented in [2]. The idea of operation of the sensory part of the device is similar to that proposed in the work [3]; however, due to the use of several dozen sensors instead of a pair of sensors, the device has greater possibilities by providing more data points per unit time.

## 2. Method and Device

The instantaneous volumetric flow rate $q_v$ is the volume of liquid flowing through the measuring tube per unit of time. In the proposed solution, the flow rate is determined indirectly by recording the time difference $\Delta t$ between the positions of the bubble detected by adjacent optical sensors located at the distance $\Delta x$ from each other (sensor beam axis distance), taking into account the inner cross-sectional area of the tube $S$:

$$q_v = S \frac{\Delta x}{\Delta t} \tag{1}$$

Block diagrams of the proposed device for measuring low liquid flow rates using the bubble method, together with an illustration of the liquid flow path and the bubble in the sensors module, are shown in Figure 1a–c. The measurement system includes (1) a sensors module containing optical sensors, pinch valves, and a system of tubes through which the liquid flows, (2) a microcontroller ($\mu$C) with software monitoring signals from sensors and controlling the opening and closing of valves, (3) an LCD display, (4) a personal computer (PC), (5) a set of electromagnetic relays activating the valves, (6) microcontroller power supply, and (7) valve power supply. The sensors module with valves is schematically illustrated in Figure 1b. The sensors module contains 30 digital slotted optocouplers (Sharp GP1S52VJ000F) on a printed circuit board. The sensors are spaced approximately constant distance apart, with an average center-to-center distance of 5.93 mm. In the sensors slot, there is a strip guiding the tube with the air bubble in liquid moving between the ends of the strip in alternate directions due to the use of valves switching the flow direction. Detection of the bubble by the digital optocoupler requires its calibration to indicate a high state (logical one) when the bubble is outside the sensor's beam field and a low state (logical zero) when the bubble is in the beam field. In order to increase the effectiveness of bubble detection, a guide for tubes with a liquid made of opaque material, common for all sensors, has been introduced into the sensor slots. Along the tube guide channel, holes with a diameter of about 1 mm were made in the central positions of the optocouplers slot. Calibration of measuring sensors (each separately) is carried out using multi-turn potentiometers R2 shown in Figure 1c. In experiments with water, due to the low light

absorption, the calibration process requires high precision and limited external lighting (an opaque guide separates the measuring tube from the external lighting).

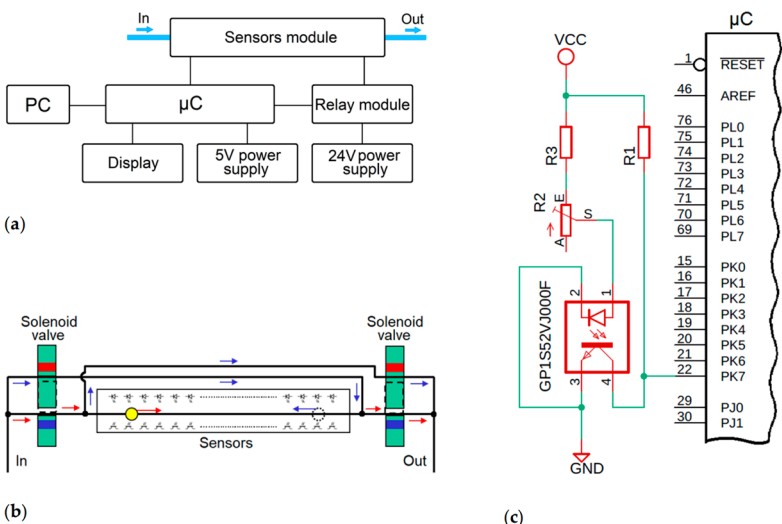

**Figure 1.** (**a**) Block diagram of the microflow meter, (**b**) sensors module containing sensors, pinch valves, and liquid tubing; the arrows show the alternating movement paths of liquid and bubble, (**c**) the method of connecting a single sensor to a microcontroller, where 1—anode, 2—cathode, 3—emitter, 4—collector.

Before starting the measurement, one should deaerate the tubes and introduce a single air bubble into the tubes with the liquid. The bubble was introduced into the measuring system by disconnecting the syringe/tube connection and applying a small amount of air into the tube. The bubble should fill the inner section of the tube, which limits the divergence of its motion in relation to the motion of the liquid. The dimension of the bubble measured along the tube should be significantly less than the distance between the detectors, so that the bubble is detected by only one sensor. At the same time, it limits the possibility of dividing the bubble into several smaller ones. Introducing more than one bubble or splitting a bubble in the flow may introduce errors in the flow rate measurement.

To ensure continuous operation of the device, it was necessary to equip the apparatus with a set of valves and their appropriate control, so that at the right moment, when the bubble approaches the end of the measuring field, the valves are switched and the direction of the liquid flow within the sensors module changes (Figure 1b). The set of valves acting as a 2-way solenoid valve was realized by using two pinch valves (HSE, Hyo ShinElectronics Ltd., Seoul, Republic of Korea). Closing the flow in such a valve consists of clamping the measuring tube in a way that completely prevents the flow of liquid. Due to the use of pinch valves, the liquid in the tubes does not have direct contact with the valves and flows in a closed system. It is worth noting that in the solution shown in Figure 1b, the length of the tubes, and thus also the volume of liquid which flows in different directions, is not identical. The effects of this asymmetry will be discussed in the section presenting the results. The operation of the device is managed by a microcontroller (Arduino Mega), whose program is written in the IDE environment. The same program determines instantaneous flow rates qv and average flow rates $q_m$ in single complete measurement cycles. Figure 2 shows the operation algorithm of the measurement loop of the program. Before starting the program, the operator should make sure that the bubble is within the sensors module.

At the very beginning, the program (part of the algorithm before the measurement loop) detects the position of the bubble on 1 of the 30 sensors and depending on whether it is detected by one of the first 15 sensors (counting from the left end of the sensors module) or one of the other sensors, the flow takes place in the right (see Figure 1b) or left direction. The movement of bubbles is detected in the measuring loop and the time is recorded. The registration of bubble detection time by the sensors is necessary to determine the

instantaneous flow rates. A new measurement cycle starts when the bubble reaches one of the two extreme positions. From that moment, the program forces a change in the flow direction and records the bubble detection times by subsequent sensors. After reaching the second extreme sensor, the flow direction changes again and another measurement cycle begins. The recording process lasts until the operator interrupts the measurement. The average flow rate $q_m$ is determined for each cycle.

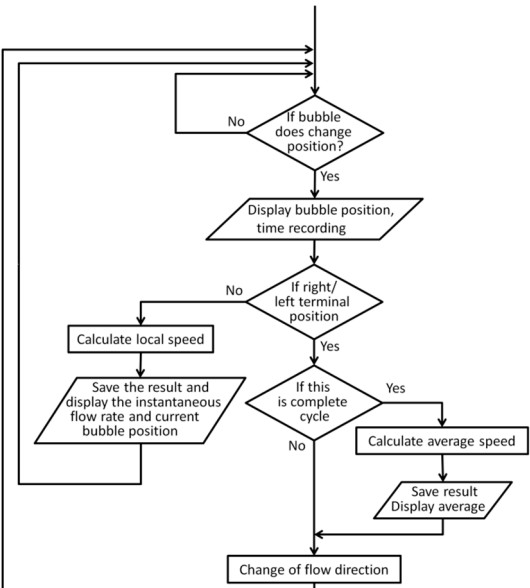

**Figure 2.** The measurement loop algorithm of the microcontroller program that controls the device and determines the flow rate.

The instantaneous flow rates, and the average flow rates after each completed cycle, are displayed on the microflow meter display. Figure 3 shows a photograph of the apparatus with visible valves, sensors module, tubing system, and display. The user can observe the flow rate value on the LCD display, and connecting the device to a personal computer enables recording the values for further processing.

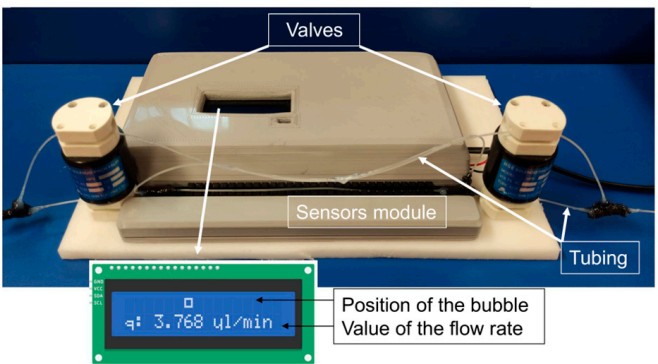

**Figure 3.** Photo of microflow meter with enlarged display.

The full research procedure shown in Figure 4 consists of the flow generator calibration (path 1 in Figure 4), determination of the inner diameter of the tubing (path 3 + 2 in Figure 4), and then the actual testing of the device (path 3 in Figure 4).

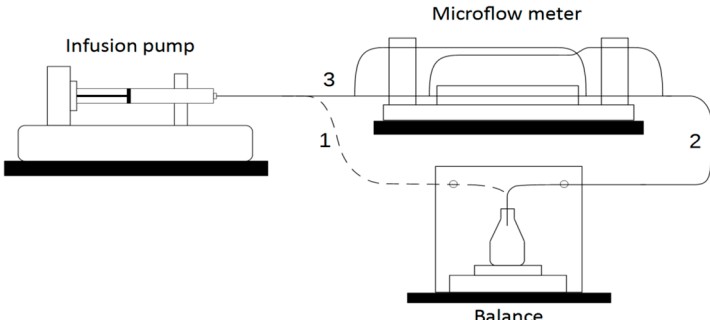

**Figure 4.** The full test procedure: (1) calibration of the flow generator, (2) determination of the inner diameter of the tube, and (3) testing of the microflow meter.

In order to calibrate the syringe infusion pump (Harvard Apparatus, Model 901), which has a 12-position gearbox and is the reference flow generator, the tests were carried out using the gravimetric method (precision balance model AJ-4200CE, Vibra, Shinko Denshi Co., Tokio, Japan). The tests were made three times for a period of about 30 min. The flow rate of the infusion pump was set to gear position 6. The obtained data were averaged and the flow rate was calculated to be 75.616 µL/min. This result was treated as the basis for calculating the reference flow rate for all pump gears, taking into account the manufacturer's data on the ratio of gears. The same result of flow rate measured by gravimetric method was used to determine the effective inner diameter of the measuring tube according to Equation (1), assuming a perfect circular inner cross-section and taking into account the result of measurement of the averaged bubble velocity ($\Delta x/\Delta t$) obtained from the microflow meter. The obtained value of the effective tubing diameter was 0.792 mm (the inner diameter declared by the manufacturer is 0.76 mm). The microflow meter tests were carried out at an ambient temperature of 20 $\pm$ 1 °C, using demineralized water supplied by a syringe infusion pump. The tubes used in the tests, which supply and discharge liquid and are located in the sensors module with pinch valves, are made of polyethylene (PE60, Becton Dickinson and Company, Franklin Lakes, NJ, USA) and have constant inner and outer diameters, according to the manufacturer's declaration, equal to 0.76 mm and 1.22 mm, respectively. Due to the sensitivity of the detectors used, it is important that the thickness of the tube wall is small, and that the tube itself hinders the detection of the bubble as little as possible (light absorption in the tube affects the transmission characteristics of the beam). Different flow rates were achieved by changing the gear ratio of the infusion pump. The transmission has 12 gears and the flow range of microliters per minute, for given tubing, provides 9 consecutive gears from 12 to 4, with gear 12 being the slowest. Specifying higher flow rates than 500 µL/min caused measurement problems, including bubble division in the tubes. Taking into account the fact that the maximum measurement time for the lowest flow rate is equal to approx. 3 h, two measurement cycles were used for the lowest flow rate in order to limit the role of slow changing external factors (e.g., temperature changes). The number of cycles increased with the flow rate.

## 3. Results

The tests with the microflow meter were carried out in the range of volumetric flow rates from less than one µL/min to several hundred µL/min. Figure 5 shows an exemplary recording of the instantaneous flow rate measurement results for the gear position 10, which gives a flow rate of 3.781 µL/min.

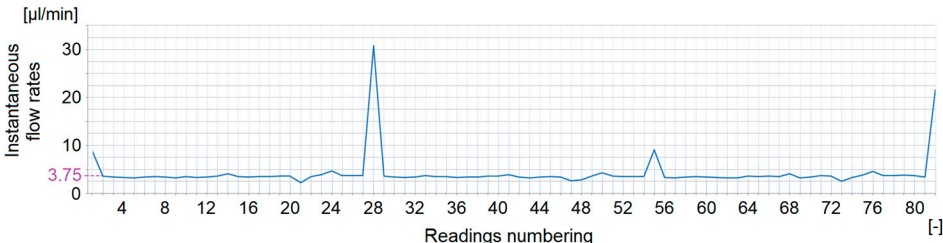

**Figure 5.** Recording of the instantaneous flow rate for the pump setting of 3.781 μL/min (numbering of consecutive readings does not include refreshing in a new cycle).

In the measurement record shown, cyclic spikes of alternating larger and smaller amplitudes can be seen (e.g., readings at positions 28 and 55, respectively). Flow rate spikes occur when the liquid flow changes direction and is caused by the acceleration of the liquid caused by the compression of the tubing in the pinch valves. The differences in the peak values can be justified by the differences in the resultant flexibility of the circuits with liquid during flows in both directions, related to the asymmetry of the length of the pipes visible in Figure 1b. A greater value of the peak amplitude occurs when there is a change in the flow direction to a direction for which the length of the tubes (volume of liquid in the tubes) is smaller. Changes in the flow direction and the associated amplitude peaks are local, non-stationary processes in time and, due to the appropriate windowing of the measurement signals, they are omitted in the calculations of instantaneous and average flow rates and standard deviations of time and distance increments.

Table 1 summarizes the results for all infusion pump flow rates tested. The average flow rates $\overline{q}$ obtained from microflow meter for $N$ consecutive complete measurement cycles (bubble passages through the sensors module) were compared with the values of the reference flow rates $q_{ref}$ (determined from the calibrated pump). The average flow rates are calculated from the averages for complete cycles $q_m$, i.e., $\overline{q} = \frac{1}{N} \sum_{m=1}^{N} q_m$. Quantitative comparison of flow rate measurement results $\overline{q}$ against the reference value $q_{ref}$ was made on the basis of the relative change.

$$\overline{\Delta} = \frac{\overline{q} - q_{ref}}{q_{ref}} 100\% \tag{2}$$

**Table 1.** Comparison of the averaged flow rate measured by the microflow meter and reference flow rate $q_{ref}$; the relative changes and estimated uncertainty of the instantaneous flow rate are shown along with the averaged time difference.

| Gear Position (Infusion Pump) | Reference Flow Rate $q_{ref}$ [μL/min] | Measured Averaged Flow Rate $\overline{q}$ [~μL/min] | Relative Change $\overline{\Delta}$ [%] | Estimated Uncertainty $u$ of Instantaneous Flow Rate [~μL/min] | Percentage Uncertainty of Instantaneous Flow Rate [%] | Averaged Time Difference $\overline{\Delta t}$ (Refresh Time) [s] |
|---|---|---|---|---|---|---|
| 4 | 378.080 | 379.388 | −0.346 | 27 | 7.14 | 0.46748 |
| 5 | 189.040 | 189.266 | −0.120 | 7.9 | 4.17 | 0.93726 |
| 6 | 75.616 | 75.531 | 0.112 | 5.7 | 7.48 | 2.34810 |
| 7 | 37.808 | 37.709 | 0.262 | 2.4 | 6.30 | 4.70325 |
| 8 | 18.904 | 18.671 | 1.233 | 1.1 | 5.94 | 9.49894 |
| 9 | 7.562 | 7.490 | 0.952 | 0.29 | 3.96 | 23.67887 |
| 10 | 3.781 | 3.766 | 0.397 | 0.20 | 5.31 | 46.96894 |
| 11 | 1.890 | 1.917 | −1.429 | 0.11 | 5.87 | 92.51681 |
| 12 | 0.756 | 0.761 | −0.661 | 0.070 | 9.92 | 233.0548 |

The uncertainty $u(q)$ was calculated for each reference flow rate according to the rule of propagation uncertainty for indirect measurements (Equation (1)), assuming independence of measurements of time difference $\Delta t$ and $\Delta x$ sensors' distance following the following formula:

$$u(q) = S\bar{q}\sqrt{\left[\frac{u(\Delta t)}{\overline{\Delta t}}\right]^2 + \left[\frac{u(\Delta x)}{\overline{\Delta x}}\right]^2} \tag{3}$$

where $\bar{q}$, $\overline{\Delta t}$ and $\overline{\Delta x}$ are the average values for given reference flow rate and $u(\Delta t)$ and $u(\Delta x)$ are uncertainties of measurements of $\Delta t$ and $\Delta x$ determined from corresponding standard deviations.

The values of $\Delta t$ and $\Delta x$ are determined from records delivered by the microflow meter and by measurements gap widths by a feeler gauge, respectively. The number of data points per cycle to determine averages and uncertainties is two fewer than the number of sensors to eliminate errors associated with spikes observed at the extreme positions of the sensor module.

As can be seen from Table 1, the discrepancies of the obtained averaged flow rate in relation to the reference values do not exceed 2%. Uncertainty of measurement of instantaneous flow rate amounts between 4 and 10%, depending on the range flow rate. The last column of Table 1 gives the average time differences $\overline{\Delta t}$ measured in bubble motion between neighboring sensors. Assuming that the flow rate measurement is based on two ycles of the bubble passing through the sensor module, we find that the total measurement time for a given pump run is approximately 56 $\overline{\Delta t}$ (30 sensors detecting the bubble were used, and the flow direction is switched after the bubble is detected by the penultimate sensor). Figure 6 shows a two-logarithmic plot comparing the flow rate measured with a microflow meter and set on the infusion pump. The course of the line connecting the measurement points does not deviate from a straight line, which is confirmed by the values of relative changes, also shown in the figure.

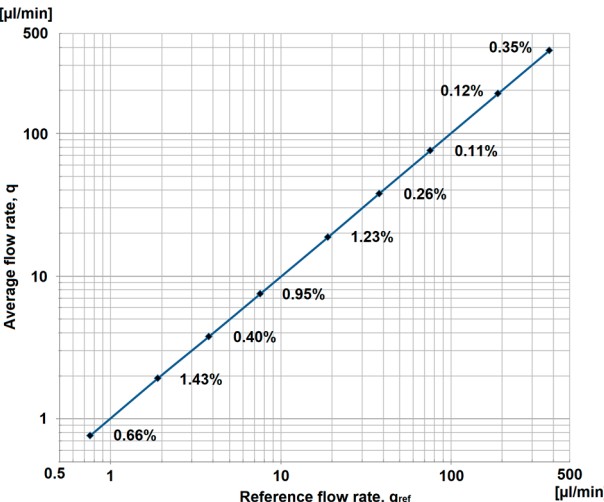

**Figure 6.** Comparison of the average flow rate $\bar{q}$ read from the microflow meter and the reference flow rate $q_{ref}$ with the marked absolute values of the relative changes $\overline{\Delta}$.

Bearing in mind the fact that the change of the flow direction is associated with different lengths of conduits, and consequently different liquid flow resistances, Table 2 compares the results of the average flow rates determined from two consecutive measurement cycles. The relative change of the flow rate in each of the flow directions in relation to the reference values $\Delta$ was also determined using Equation (2), where $\bar{q}$ was replaced by $q_m$. The results of the average flow rate measurements obtained for two consecutive cycles (for different flow directions) in most cases do not differ from the reference value by more than 1%.

**Table 2.** Comparison of average flow rates from two consecutive cycles (different direction of bubble movement) for all reference flow rates and relative flow rate changes.

| Average Flow Rate for a Single Cycle $q_m$ [μL/min] | Relative Changes Δ [%] | The Direction of Movement of the Bubble |
|---|---|---|
| Reference flow rate $q_{ref}$ = 0.756 [μL/min] | | |
| 0.749 | −0.90 | Right (=>) |
| 0.754 | −0.29 | Left (<=) |
| Reference flow rate $q_{ref}$ = 1.890 [μL/min] | | |
| 1.960 | 4.67 | => |
| 1.860 | −1.58 | <= |
| Reference flow rate $q_{ref}$ = 3.781 [μL/min] | | |
| 3.730 | −1.35 | => |
| 3.803 | 0.60 | <= |
| Reference flow rate $q_{ref}$ = 7.562 [μL/min] | | |
| 7.486 | −1.00 | => |
| 7.507 | −0.72 | <= |
| Reference flow rate $q_{ref}$ = 18.904 [μL/min] | | |
| 18.702 | −1.07 | => |
| 18.496 | −2.16 | <= |
| Reference flow rate $q_{ref}$ = 37.808 [μL/min] | | |
| 37.443 | −0.97 | => |
| 37.617 | −0.51 | <= |
| Reference flow rate $q_{ref}$ = 75.616 [μL/min] | | |
| 75.158 | −0.61 | => |
| 75.502 | −0.15 | <= |
| Reference flow rate $q_{ref}$ = 189.04 [μL/min] | | |
| 187.72 | −0.70 | => |
| 188.26 | −0.41 | <= |
| Reference flow rate $q_{ref}$ = 378.08 [μL/min] | | |
| 377.51 | −0.15 | => |
| 381.43 | 0.88 | <= |

## 4. Discussion

The proposed microflow meter solution enables long-term measurements and observations of flow rate changes in the order of microliters per minute. The flow rate limits can easily be changed by using tubing with a different inner diameter, bearing in mind that the change will be proportional to the square of the diameter. The only element requiring replacement, apart from the tube, is the tube guide located in the sensor slot. The upper limit of the volumetric flow rate used is related to the observation that at higher flow rates, the bubbles split, especially when changing the direction of the flow. Above 0.5 mL/min, the device does not measure correctly, and this is a situation when the flow rate in the tube is approx. 14 mm/s. The lower limit is related to the lowest pump flow. It is worth emphasizing that in the entire measuring range, the Reynolds number, which determines the ratio of inertial forces to viscous forces in the flow, does not exceed the value of 10, which means that the flow is always laminar.

The solution of the microflow meter allows for measurements in which the flow takes place in a closed system of tubes with liquid, without contact with other elements of the apparatus. The only interference in the liquid is the introduction of a bubble, which, however, remains within the sensors module throughout the measurement. Appropriate ISO 13485-certified tubing should be used when measuring bodily fluids in a living organism, and bubble insertion should be performed under aseptic conditions.

The tested device can be used to measure instantaneous flow rates of variable intensity over time. Due to the constant distance between the sensors, the refresh time of instantaneous volumetric flow rate measurements is inversely proportional to this flow rate. In the case of measuring the average flow rate, it is worth remembering that increasing the number of measurement cycles is associated with a longer averaging period, and the final results will contain fewer measurement records. In addition, with longer measurements, a greater influence of slow changing external factors, such as temperature, is possible. In any case, the length of the measurement periods should be determined on a compromise basis.

The presented measurements were made for demineralized water using a specific type, length, and diameter of the tube through which the liquid flows in the sensor gap. The use of a different liquid or a different tube requires recalibration of the sensors. For liquids that strongly absorb IR light in the sensor gap, the contrast of the bubble compared to the liquid, in the given case of the tube material, may be higher, which should result in higher quality (precision) measurements.

A measurement procedure may contain several sources of error. The microcontroller program stops the measurements when the sensors do not detect a bubble for a certain period of time or when a bubble is detected by several sensors at the same time (this program function is not shown in the algorithm presented in Figure 2). These problems can be caused by incorrectly set thresholds for detecting bubbles by the sensors, division of the bubble into several smaller ones, or the introduction of a bubble that is too long. The experience gained from the tests of the designed microflow meter shows that it is advantageous if the length of the bubble does not exceed several tubing diameters. In this case, the program written for the microcontroller can easily handle the unambiguous detection of the bubble and the determination of its speed.

Small differences in the distance between adjacent sensors resulting from the imperfect way of mounting the optocouplers on the PCB and the uncertainty of the measurement of the air bubble transit time between the sensors affect the instantaneous flow rate, and the percentage uncertainty is from approx. 4 to 10% depending on the measurement range (Table 1). The measurement uncertainty of the average value of the volumetric flow rate determined for at least two measurement cycles (each covers the entire length of the sensors module) decreases accordingly to the number of averaging.

Due to the fact that the tubes used in the measurement system are several dozen centimeters long, it should be taken into account that flow resistance through the tubes may affect the measurement results. In the case of using an infusion pump, these resistances do not play a major role. This is confirmed by the comparison of the results of average flow rates determined from subsequent single cycles (Table 2), for which the tube lengths differ significantly.

Comparing the instantaneous volumetric flow rates for the constant reference flow rate (see, e.g., Figure 4), it can be seen that in the second half of each cycle, there are greater flow rate fluctuations than in the first half, regardless of the direction of flow. The work carried out did not find an explanation for this effect. It is worth emphasizing, however, that the mentioned flow rate fluctuations do not contribute significantly to the averages calculated from the measurement cycles.

Evaluation of the quality of the results of measurements of average volumetric flow rates from the microflow meter in this work is based on a comparison with the reference flows set from the syringe flow pump (Table 1) using the parameter of relative changes. The choice of the parameter results from the fact that sources of inaccuracies may lie both on the side of the microflow meter and the infusion pump [13]. It is worth noting, however,

that the infusion pump was previously calibrated using the gravimetric method. As a result, the microflow meter testing procedure used was significantly shorter compared to the procedure in which reference volumetric flow rates for all ranges would come from the gravimetric method.

Taking into account at least two measurement cycles to determine the average flow rate, the relative changes in the proposed solution do not exceed 2%. In earlier studies of a device using the bubble method, it was shown [2] that the measurement uncertainty did not exceed 4%. A comparison of the averages from individual subsequent measurement cycles (Table 2) showed that in most cases, the relative changes in the flow rates are at the level of 1%; however, in a single case, they reached the level of 4.7%. Such differences may result from errors in determining the flow rate, e.g., associated with the division of bubbles. Due to the fact that the movement of the bubble takes place in a fragment of the pipe hidden in the sensors module, this hypothesis has not been confirmed.

In the literature, one can find papers showing that the velocity of bubble motion and the average velocity of the liquid in the capillary can be different, see, e.g., [14,15]. Most studies are concerned with capillary numbers between $10^{-3}$ and 1, and then in circular capillaries, for pure water, the bubble velocity is greater than the average liquid velocity, whereas the presence of surfactants results in a lower bubble velocity than the average liquid velocity. In the present study, the effective diameter of the tubes was determined by comparing the result of the bubble and gravimetric methods. This solution eliminates the problem of having to take into account the bubble and liquid velocity discrepancies. One problem with the current measuring system is that it is not possible to observe the movement of the bubble during measurement. The measuring tube guide protects the capillary from ambient light, but thus affects the inability to detect possible bubble problems during measurement. The presence in the measurement results of non-stationary components at the end positions of the bubble in the sensor module is due to the operation of the pinch valves used. Replacing these valves with another type that does not cause a sudden increase in fluid pressure and flow rate jump would reduce this effect, although it would limit the medical applications of the developed microflow meter.

## 5. Conclusions

A simple device for precise measurement of the slow volumetric flow rate of liquid is described. The device allows for the measurement of instantaneous and average values. It can be carried out for any length of time in a closed system, not counting the measuring tube. Optical sensors were used to track the movement of the bubble moving in the liquid tube. Due to the use of an appropriate system of valves switching the flow direction during the measurement, the bubble tracked by the sensors remains present in the sensors module all the time. The results of the conducted tests were presented in the range of volumetric flow rates in the order of microliters per minute using a syringe infusion pump as a source forcing the flow. Differences in the flow rate measured with a microflow meter in relation to the reference flow rate were assessed on the basis of averages covering at least two measurement cycles (flow in the sensors module in both directions). In addition, the flow rates from two consecutive measurement cycles were compared. A good agreement of the measurement results with the reference flow rates and no systematic dependence on the flow direction in the sensors module were found. Factors affecting the measurement results were discussed and directions for further improvement of the method were indicated.

Considering the further development of the method, it is worth considering the use of analog optical sensors instead of digital ones. Such a solution should enable software auto-calibration of the sensors in order to determine the bubble detection threshold. This would be a particularly useful feature when using different liquids or tubes. Then, it could also be considered to adapt the device to measure low gas flow rates, where instead of a bubble, the tracked object would be a drop of liquid not wetting the tube wall. Another modification of the device could be the use of a symmetrical solution in terms of the length of the tubes through which the liquid flows in subsequent measurement cycles. This can

be accomplished by a different arrangement of pinch valves, and this solution could be important in cases where flow resistance through liquid lines affects the precision of the measurement process. For applications that do not require a closed system for the fluid, it would be worth considering the use of other less interfering valves that could reduce flow rate jumps with changes in flow direction. In addition, it is worth automating the procedure for introducing the bubble into the system and providing for a bubble size test on the results for particular flow velocities.

**Author Contributions:** Conceptualization, B.S. and M.K.; methodology, B.S. and M.K.; software, M.R.; validation, M.R. and M.K.; investigation, M.R.; resources, M.K.; writing—original draft preparation, M.K. All authors have read and agreed to the published version of the manuscript.

**Funding:** This research received no external funding.

**Data Availability Statement:** The data presented in this study are available in this article.

**Conflicts of Interest:** The authors declare no conflicts of interest.

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
