# Peer review of "Optical Bubble Microflow Meter for Continuous Measurements in a Closed System"

_electronics, doi:10.3390/electronics13051000_

Round 1

Reviewer 1 Report

Comments and Suggestions for Authors

The design, operation and test results of a simple microprocessor-based device for measuring slow liquid flows has been proposed and demonstrated in this work. And experimental tests have been carried out to verify its effectiveness. In general, the paper is well-written and the idea is clearly articulated. I suggest that with some minor modifications, the paper can be published. Here are some suggestions:

1. The time consumption of the proposed microprocessor-based device for measuring slow liquid flows should also be analyzed.

2. In the experimental results, a comparison between the proposed microprocessor-based device and the traditional measuring devices should be supplemented.

3. The limitations of the research work should also be discussed.

4. In conclusion, the future research work should also be given.

Comments on the Quality of English Language

The English in this manuscript should be further refined.

Reviewer 2 Report

Comments and Suggestions for Authors

This manuscript proposed a micro flow measurement scheme using bubble movement in a capillary tube and conducted experimental verification. But there are still some issues that need further description:
1 How to generate a suitable bubble for measurement in a capillary tube
2 When switching the direction of bubble movement, measurement errors usually occur. How large is this error and how can it be suppressed.
3 The use of “closed loop” in the title of the manuscript may not be appropriate

Reviewer 3 Report

Comments and Suggestions for Authors

The paper describes a device that uses the transit of a bubble in a capillary tube to measure flow rate. The velocity of the bubble is tracked with optical sensors and the flow is reversed to keep the bubble in the flow loop. The flow measurements are calibrated with a gravimetric method and a syringe pump is used to deliver flow to the system. The method is interesting and the implementation appears to be good. My main concern is about the relative velocity of the bubble to the mean velocity of the fluid.

The difference in the velocity of a bubble or drop to the mean velocity of the carrier fluid needs to be examined. This will create a bias in the measurement accuracy that scales with Capillary number (delta V ~ Ca^(2/3)). Please see review by Baroud et al (2010) for a brief section on the topic and Ratulowski & Chang (1989) for the flow of a bubble. A discussion of this affect and its impact on the measurement should be included in a revision.

The magnitude of this difference should be estimated using the parameters from the experimental system. A measurement of the velocity difference between the mean fluid velocity and the bubble velocity would be better, but a more accurate measurement of the tubing diameter would be needed.

The difference in bubble velocity to the mean velocity may be compensated for by using the bubble velocity to calibrate the diameter of the tubing. The measured size of the tubing is slightly larger than the manufacturer’s value. Which might be a result of the bubble traveling faster than the mean velocity of the fluid. A different independent measurement of the tubing diameter would be warranted.

The uncertainties are taken from the standard deviations of the time and distance measurements. I assume this does not include the spikes shown in Fig 5?

The relative changes shown in Figure 6 are all positive. Why are these values different than the ones shown in Table 1?

Comments on the Quality of English Language

The writing is satisfactory. Some typos and minor errors should be corrected.

Round 2

Reviewer 2 Report

Comments and Suggestions for Authors

The author made revisions to the manuscript according  the review comments.

Comments on the Quality of English Language

N/A

Reviewer 3 Report

Comments and Suggestions for Authors

The inclusion of discussion of some of the uncertainties around the velocity of the bubble have improved the paper. I recommend publication.

Compliance under pressure might be the origin of the the larger effective diameter since thin wall tubing is used. Possibly consider using glass capillaries for future developments.

Comments on the Quality of English Language

The English is fine.